# Correlates of and changes in aerobic physical activity and strength training before and after the onset of COVID-19 pandemic in the UK: findings from the HEBECO study

Aleksandra Herbec ![ORCID] ,[1,2] Verena Schneider,[1] Abigail Fisher,[1] Dimitra Kale,[1] Lion Shahab,[1] Phillippa Lally[1]

[1]Department of Behavioural Science and Health, University College London, London, UK
[2]Institute - European Observatory of Health Inequalities, Calisia University, Kalisz, Poland

**Correspondence to**
Dr Aleksandra Herbec;
a.herbec@ucl.ac.uk

## ABSTRACT

**Objectives** Understanding changes in moderate to vigorous aerobic physical activity (MVPA) and muscle-strengthening activity (MSA) at the start of the COVID-19 pandemic and their correlates (socio-demographics, health characteristics, living and exercise conditions and pre-pandemic MVPA/MSA) can inform interventions.

**Design** A cross-sectional analysis of retrospective and concurrent data on MVPA/MSA.

**Setting** An online survey in the UK.

**Participants** 2657 adults (weighted n=2442, 53.6% women) participating in the baseline survey (29 April 2020–14 June 2020) of the HEalth BEhaviours during the COVID-19 pandemic (HEBECO) study.

**Primary and secondary outcome measures** Meeting WHO-recommended levels for MVPA/MSA/both (vs meeting neither) during the first lockdown and changes in MVPA/MSA from before to since the COVID-19 pandemic following stratification for pre-pandemic MVPA/MSA.

**Results** A third of adults maintained (30.4%), decreased (36.2%) or increased (33.4%) MVPA. For MSA, the percentages were 61.6%, 18.2% and 20.2%, respectively. MVPA increased or decreased by an average of 150 min/week and 219 min/week, respectively, and MSA by 2 days/week. Meeting both MSA+MVPA recommendations since COVID-19 (vs meeting neither) was positively associated with meeting MVPA+MSA before COVID-19 (adjusted OR (aOR)=16.11, 95% CI 11.24 to 23.07) and education: post-16 years of age (aOR=1.57, 95% CI 1.14 to 2.17), and negatively associated with having obesity (aOR=0.49, 95% CI 0.33 to 0.73), older age (65+ years vs ≤34 years; aOR=0.53, 95% CI 0.32 to 0.87) and annual household income of <50 000 GBP (aOR=0.65, 95% CI 0.46 to 0.91). The odds for decreasing MVPA were lower for white ethnicity (aOR=0.62, 95% CI 0.44 to 0.86), education: post-16 years of age (aOR=0.73, 95% CI 0.58 to 0.91) and access to garden/balcony (aOR=0.75, 95% CI 0.60 to 0.94), and were higher for those living in total isolation (aOR=3.81, 95% CI 2.33 to 6.23), with deteriorated psychological well-being (aOR=1.40, 95% CI 1.15 to 1.71) and conditions limiting physical activity (aOR=1.74, 95% CI 1.27 to 2.39). The odds for decreasing MSA were higher for having overweight (aOR=1.88, 95% CI 1.39 to

## Strengths and limitations of this study

⇒ The study presents novel data on the unique correlates of increases and decreases in WHO recommended levels of physical activity at the onset of the COVID-19 pandemic.
⇒ Two types of physical activity, moderate-to-vigorous aerobic physical activity and muscle strength training are assessed in the study.
⇒ It was a cross-sectional and self-reported study among a convenience sample.
⇒ Biases were minimised by weighing data using UK Census data.
⇒ We tested robustness of findings in sensitivity analyses.

2.55), obesity (aOR=23.38, 95% CI 2.23 to 5.14) and being employed (aOR=1.81, 95% CI 1.34 to 2.46).

**Conclusion** Aerobic and strength training were differently impacted during the first UK lockdown, with poorer outcomes associated with older age, lower education and higher body mass index. Targeted interventions may be required to avoid pandemic-related inequities in physical activity.

## INTRODUCTION

Physical activity has been negatively affected by the COVID-19 pandemic, which could have major implications for general health and COVID-19 outcomes.[1–3] As the COVID-19 pandemic continues and new restrictions may be introduced with new waves of the pandemic, it is paramount to understand how physical activity, and especially aerobic and muscle-strength training, were affected during the early stages of the pandemic, and which groups may require most support.

Adults who engage in less than 30 min of moderate physical activity per week are considered to be inactive.[4 5] In the UK[6] and other

countries (eg, the US[7] guidelines for physical activity), frequency and duration are complex, and often vary by age or clinical profile of the target population. However, a consensus exists that for the best health outcomes adults are recommended to engage in moderate-to-vigorous intensity aerobic physical activity (MVPA; ie, activities that increase heart rate and make one feel warmer) for at least 150 min/week, as well as in muscle-strengthening activity (MSA; eg, strength/resistance training) for at least 2 days/week.[8] MVPA and MSA lower morbidity and mortality both independently and combined.[9–14] Importantly, improving MVPA or MSA requires different strategies and interventions at both population and individual level, thus both are important outcomes to research.[15]

The COVID-19 pandemic[15] and social distancing measures introduced in the UK and other countries limited many opportunities to engage in physical activity.[15 16] The first national lockdown in the UK was introduced on 23 March 2020. As part of the lockdown measures, sports centres and gyms were temporarily closed and team sports banned, while access to outdoors spaces (eg, remote national parks) was greatly reduced (ie, some parks were closed or were only available to local visitors). By contrast, certain lockdown measures introduced in the UK could enable exercising, including the introduction of remote working that offered greater flexibility in scheduling and engaging in exercising at home or in local parks.[15] Furthermore, exercising outdoors was listed as one of the few activities that were still permitted even during the strictest lockdown in the UK, alongside shopping for essential items or medicine and going to work.

A large systematic review[17] showed that the majority of studies published by October 2020 reported declines in different measures of physical activity and increases in sedentary behaviour during the initial stages of COVID-19 pandemic.[17 18] However, some studies found that a small minority of adults increased their activity,[19] suggesting it is important to identify factors associated with both the detrimental and beneficial changes to activity levels. Female gender, lower income, older age, health conditions, perceived risks, poorer mental health and non-white ethnicity[19–25] were all associated with lower physical activity levels. However, few studies accounted for other factors that could be relevant during the pandemic, such as having access to indoor and outdoor exercising space or household makeup.[19]

Furthermore, research on how the COVID-19 pandemic has affected physical activity to date has often focused on moderate-to-vigorous physical activity (MVPA), without consideration of MSA, or failed to distinguish between MVPA and MSA,[17 19 26 27] or assessed only perceived, qualitative changes to physical activity levels with respect to pre-pandemic levels (eg, participants were asked to report if they exercised 'more' or 'less' since before the pandemic started rather than providing detailed information about MVPA and MSA responses).[28 29] Moreover, few studies that assessed and reported declines in MVPA and

MSA, for example, among adults in Russia[30] and in Italy,[31] did not report on factors associated with these specific changes.

In order to identify groups in the population that may require targeted physical activity interventions during future periods of lockdowns or restrictions to physical activity, it is important to research the patterns and factors associated with changes in levels of both MVPA and MSA that are recommended by the WHO.

## Aims

We characterised separately and combined MVPA and MSA levels during the periods before and since the COVID-19 pandemic had started in the UK, with the latter covering the period of the first strictest UK lockdown up until 14 June 2020. The research questions (RQs) were:

1. RQ1: what were the levels and changes in MVPA and MSA among UK adults from before to since COVID-19?
2. What socio-demographic, environmental and health factors were associated with:
   – RQ2: meeting WHO recommendations for MVPA and MSA since COVID-19?
   – RQ3: (1) decrease in MVPA among those who were active and (2) increase in MVPA among those who were inactive from before to since COVID-19?
   – RQ4: (1) decrease in MSA among those who were active and (2) increase in MSA among those who were inactive, from before to since COVID-19?

## METHODS

### Study design

The study involved cross-sectional data analysis from the baseline online survey of the HEalth BEhaviours during the COVID-19 pandemic (HEBECO) study (https://osf.io/sbgru/). Data were collected using REDCap software[32] at UCL. The protocol for the present data analysis was preregistered (https://osf.io/ejghs/). Departures from the protocol are explained in online supplemental materials 1. The results from all the preregistered analyses not reported here are available elsewhere (https://osfio/2ujxq/). The reporting follows the Strengthening the Reporting of Observational Studies in Epidemiology guidelines.[33]

### Participants and procedures

Recruitment involved a UK-wide campaign, including paid and unpaid posts, on social media and information shared through the networks of Cancer Research UK, Public Health England, other charities, sports clubs, universities and local authorities. Interested participants were directed to the study website where they were shown a participant information sheet, provided informed consent and then completed the online survey. Participants with complete data on the measures of interest for this study, and who enrolled between 5 May 2020 (start of the data collection for all study variables, during the first UK lockdown) until 14 June 2020 (the last day prior

to easing of restrictions of the first UK lockdown) were included in the analytical sample.

### Patient and public involvement

Patients or members of the public were not involved in the HEBECO baseline design, but the survey was shared with researchers at CRUK and Public Health England's behavioural insights function in order to collect their feedback and suggestions for questions and answers wording, where relevant (eg, when questions were not validated). Participants in the HEBECO study were encouraged to leave comments about specific behaviours and the survey itself, and these were incorporated into subsequent follow-up surveys (not the present study), where possible and relevant.

### Measures

For the wording of all measures, see https://osf.io/bja7g/ and online supplemental materials 2. At baseline, participants indicated the time when their lives started to be affected by the COVID-19 pandemic in any way, which was used as an anchor to assess before/since the COVID-19 pandemic onset MVPA and MSA. Almost half (44.0%, weighted) of adults selected the second half of March, followed by 26.9% who selected the first half of March.

### Meeting WHO recommendations for MVPA and MSA

The before/since COVID-19 MVPA and MSA were measured using the following questions.[14 34] For MVPA, participants were first asked, 'In the month before/since COVID-19, on average, how many times per week did you do at least 15 min or more of MVPA? Examples: brisk walk, jogging, dancing, cycling for recreation or commute, swimming, team or racket sports'. The answers were capped at 14+ "times" per week (ie, sessions), equivalent to at least two times (or sessions) of MVPA per day. The "15-minute or more" for the MVPA period was selected as it corresponds to the minimum amount of time of PA needed for mortality reduction and extending lifespan.[35] Those who engaged in at least one such session were asked, 'In the month before/since COVID-19, how long (in minutes) was your average session of MVPA? Do not include strength training'. Total weekly MVPA was computed by multiplying the session number by the minutes.

MSA was assessed by a single question: 'Before/since COVID-19, on average, how many days per week did you do strength training? Examples: pilates, push-ups, squats, yoga and exercises involving free weights, weight machines or elastic band'. The answers were capped at 4+ (days per week), given that the recommendations are for 2 days of MSA per week.

Data on MVPA and MSA were used to categorise participants, separately for before/since COVID-19, into four groups based on whether they met the WHO recommendations for MSA (at least 2 days/week) and MVPA (≥150 min/week)[8]: meeting neither, meeting MSA only, meeting MVPA only or meeting recommendations for both MSA and MVPA.

### Change in MVPA and MSA

Changes in MVPA (RQ3) and MSA (RQ4) were calculated by subtracting the before COVID-19 levels of MSA or MVPA from the since COVID-19 levels. Participants were categorised into: maintenance (change in MVPA <20 min; change in MSA=0), decrease or increase in activity (MVPA by ±≥20 min; MSA by ±≥1 day). Dichotomous variables were then created: decreased (vs maintained/increased) and increased (vs maintained/decreased) MVPA and MSA levels.

To minimise the bias due to ceiling and floor effects when assessing changes to MVPA and MSA, the analyses for RQ3 and RQ4 were conducted after stratifying for before COVID-19 MVPA and MSA. Separately for MVPA and MSA, participants were categorised into: inactive (ie <30 min of MVPA/week[4 5]; 0 days of MSA/week) and active (≥30 min/week of MVPA; ≥1 day/week of MSA) before COVID-19.

Participants who were active since COVID-19 were also asked, 'Do you do the same form of exercise as you did before the COVID-19 (even if it is in a different location)?'. The answer options were: none of the same/some of the same/about half the same/mostly the same/exactly the same.

### Explanatory factors and correlates

Socio-demographic characteristics and living conditions assessed were: gender (participants were asked to select which answer they identify with most, and the answers were dichotomised into: female/all other), ethnicity (white/non-white); education (post-16-years of age/other); employed (yes/no); furloughed/laid off (yes/no); income (low-middle <50 000 GBP/high ≥50 000 GBP/prefer not to say) and age (in decades was entered as a continuous variable where it met assumptions, or as a 3-level categorical variable: ≤34 years, 35–64 years and 65+ years).

We assessed health behaviours, health and living conditions that could impact on MVPA or MSA levels: self-reports of any condition that limited physical activity (yes/no); body mass index (BMI[34]: normal and underweight ≤24.99 $kg/m^2$/overweight=25–29.99 $kg/m^2$/obese ≥30 $kg/m^2$); smoking status (current smoker vs not)[36]; weekly frequency of alcohol drinking[37]; deterioration in psychological well-being from before to since COVID-19 (yes/no); participants' perceived risk of COVID-19 to their health ('no or minor risk'/other); access to a garden or balcony big enough to exercise in comfortably (yes/no); access to a public park/green space that is within a walking distance and open during COVID-19 (yes/no); living with children aged ≤15 years (yes/no); living with vulnerable persons (persons over the age of 70 years, in poor health or who may be vulnerable to COVID-19 (yes/no)). Participants were asked about the status of their isolation at the time of completing the survey: 'Which

type of COVID-19 induced isolation are you experiencing?': (1) total isolation/quarantine (not leaving the house for any reasons, not even to buy groceries or medications or to exercise); (2) some isolation (not leaving the house except to buy essential items, such as groceries or medication or to exercise); (3) general isolation but still go out to work (still go out to work and to buy essential items, such as groceries or medications or to exercise); (4) no isolation (I am free to leave the house whenever I like, including participating in social gatherings or group sports, going to a bar or restaurant and travelling for leisure). The answers were dichotomised into: (1) total isolation/(2)–(4) all other.

The regression models described below included two time covariates to account for weather changes: enrolment time (up until 15 May, the second half of May and the first half June) and time when COVID-19 started to affect individuals (before mid-March/later) as these could affect exercise levels.

## Analysis

Analyses were conducted in SPSS V.26 with the data weighted using the 2018 Census and APS mid-year estimates for age, gender, ethnicity, country of living and household income. The analysis used weights trimmed to top 98th percentile to minimise the impact of extremely high weights.[38] Differences between the included and excluded participants were assessed with $\chi^2$ test for categorical and t-test for continuous data.

For RQ1, descriptive statistics were computed to characterise the levels of MVPA, MSA, inactivity before/since the pandemic and changes in exercise form. For RQ2, univariate and fully adjusted multinomial regression models were computed. For RQ3 and RQ4, the sample was stratified using before pandemic MVPA or MSA levels to assess outcomes of interest using univariate and adjusted logistic regression models. All independent variables were entered into the models together as they were selected due to their previously established or theoretical importance. Sensitivity analyses involved replicating the analyses using unweighted data to check for the robustness of the results, and for RQ3 using different cut-off values of 15 mins and 30 mins. Familywise error was corrected for by using the Benjamini-Hochberg procedure separately for each RQ.[39]

## RESULTS

Out of 2992 participants who were included in the HEBECO baseline sample, 2657 adults (weighted N=2442) had complete data for this study and were included in the analytical sample. Table 1 presents comparisons of the excluded and included sample. The included weighted sample comprised 52% females, 90.5% of white ethnicity and 67.0% with high school education or higher.

The sections below summarise the significant results for the weighted fully adjusted analyses. Results from univariable analyses are reported in online supplemental

materials 3, and from sensitivity analyses in online supplemental materials 4 and 5. The results of the sensitivity analyses did not change the direction or magnitude of the results.

### RQ1: changes in MVPA and MSA from before to since COVID-19

Table 2 presents data on physical activity levels before and since COVID-19. Before COVID-19, 17.6% of adults engaged in no MVPA and MSA, 55.3% engaged in no MSA and 19.4% engaged in no MVPA. Just under 15% of adults met the recommended levels of both MVPA and MSA before and since COVID-19. The proportion of those who had no MVPA or MSA activity increased minimally (17.6%–22.1%), which was primarily driven by declines in MVPA. Among those who were active since COVID-19, 41.5% continued with mostly or exactly the same form of activity as before COVID-19.

From before to since COVID-19, similar proportions of adults maintained (30.4%), decreased (36.2%) or increased (33.5%) their weekly MVPA. Among those who were active before COVID-19, 46.7% decreased MVPA. The decrease was of on average 219 min/week. Among those who were inactive before COVID-19, 29.7% increased MVPA, with an average increase of 144 min/week.

Maintenance of MSA was relatively more common (61.6%), with 18.2% decreasing and 20.2% increasing the number of days they engaged in MSA. Among those who were active before COVID-19 (40.2%), 45.1% decreased MSA, with an average decrease of 1.9 days/week. Among those who engaged in no MSA before COVID-19 (59.8%), 16.4% increased MSA, with an average increase of 2.5 days/week.

### RQ2: predictors of meeting WHO recommendations

Table 3 presents results from fully adjusted models. Being aged 65+ years (adjusted OR (aOR)=0.53, 95% CI 0.32 to 0.87), having a lower pre-COVID-19 household income (aOR=0.65, 95% CI 0.46 to 0.91), having a condition limiting physical activity (aOR=0.44, 95% CI 0.28 to 0.71), having obesity (aOR=0.49, 95% CI 0.33 to 0.73), living in total isolation (aOR=0.44, 95% CI 0.23 to 0.83) and deterioration in psychological well-being (aOR=0.56, 95% CI 0.43 to 0.73) were significantly associated with lower odds, while having at least high school education (aOR=1.57, 95% CI 1.14 to 2.17) or meeting the WHO recommended levels of MVPA (aOR=3.88, 95% CI 2.64 to 5.70), MSA (aOR=6.38, 95% CI 4.26 to 9.54) or both before COVID-19 (aOR=16.11, 95% CI 11.24 to 23.07), were significantly associated with greater odds of meeting both MVPA and MSA WHO guidelines since COVID-19, as compared with not meeting either.

Having a condition that limited physical activity (aOR=0.43, 95% CI 0.29 to 0.63), being in total isolation (aOR=0.26, 95% CI 0.14 to 0.50) and deterioration in psychological well-being (aOR=0.71, 95% CI 0.56 to 0.89) was significantly associated with lower odds,

**Table 1** Sample characteristics (weighted) and comparison of participants who were included and excluded from the study (due to incomplete data)

| | Included sample N=2442 | Excluded sample N=350 | P value |
|---|---|---|---|
| Female | 52.6% | 48.7% | 0.179 |
| Age (years): mean (SD) | 50.0 (16.0) | 42.7 (18.9) | <0.001 |
| White ethnicity | 90.5% | 82.4% | <0.001 |
| Household income ≥50 000 GBP | 18.1% | 15.5% | 0.095 |
| Income: <50 000 GBP | 73.5% | 73.1% | |
| Income: prefer not to say | 8.3% | 11.5% | |
| Education: high school or higher | 67.0% | 69.1% | 0.432 |
| Employed | 48.3% | 40.7% | 0.008 |
| Laid-off/furloughed | 12.7% | 15.2% | 0.197 |
| Condition limiting physical activity | 16.3% | 21.2% | 0.035 |
| BMI ≤24 kg/m$^2$ | 43.2% | 27.3% | <0.001 |
| BMI: 25–29.99 kg/m$^2$ | 35.1% | 16.8% | |
| BMI: obese: 30+ kg/m$^2$ | 21.7% | 8.4% | |
| BMI: do not know/prefer not to say* | 0% | 47.6% | |
| Total isolation | 7.8% | 8.8% | 0.541 |
| Minor/no COVID-19 risk percept | 33.3% | 38.6% | 0.069 |
| Living with children | 17.0% | 15.7% | 0.549 |
| Living with vulnerable | 15.1% | 21.8% | 0.001 |
| Access garden/balcony | 72.2% | 62.6% | <0.001 |
| Access green space | 59.5% | 52.4% | 0.012 |
| Time life affected by COVID-19 mid-March or sooner† | 44.8% | 53.3% | 0.003 |
| Enrolled from 1 June (reference) | 4.4% | 5.4% | 0.404 |
| Enrolled up until 15 May | 50.1% | 46.7% | |
| Enrolled second half of May | 45.4% | 47.9% | |
| Smoker | 24.1% | 38.9% | <0.001 |
| Weekly alcohol drinking, none (reference) | 21.8% | 21.8% | 0.920 |
| Weekly or less | 27.9% | 26.9% | |
| More than weekly | 50.3% | 51.3% | |
| Deteriorated psychological well-being | 54.1% | 52.8% | 0.708 |
| Meeting WHO MVPA and MSA levels before COVID | 15.6% | 17.3% | 0.218 |
| Meeting only MVPA before | 23.0% | 22.0% | |
| Meeting only MSA before | 13.0% | 7.7% | |
| Meeting neither before | 48.4% | 53.0% | |

*As per protocol, the participants who did not provide BMI data (eg, they selected 'prefer not to say' or 'do not know' on the question on height or weight) but who provided other study data were to be included in the analyses but were ultimately excluded due to having missing data on other variables.
†Time that participant's life started to be affected by COVID-19 in any way.
BMI, body mass index; MSA, muscle-strengthening activity; MVPA, moderate-to-vigorous aerobic physical activity.

and meeting the WHO recommended levels of MVPA (aOR=7.57, 95% CI 5.82 to 3.39) or both MVPA and MSA (aOR=2.31, 95% CI 1.59 to 3.39) was significantly associated with greater odds of meeting MVPA only since COVID-19. Being aged 35–64 years (aOR=0.56, 95% CI 0.38 to 0.82), and having obesity (BMI ≥30 kg/m$^2$; aOR=0.37, 95% CI 0.24 to 0.56), were associated with lower odds, while higher education, and meeting both MVPA and MSA or only MSA recommendations before COVID-19, were associated with greater odds of meeting the recommendations for MSA only since COVID-19.

### RQ3: associations with changes in MVPA

Older age (aOR=0.91, 95% CI 0.85 to 0.98), white ethnicity (aOR=0.62, 95% CI 0.44 to 0.86), education:

**Table 2** MVPA and MSA before and since COVID-19

| | Before COVID-19* | Since COVID-19* |
|---|---|---|
| No activity (0 sessions MVPA and 0 days MSA): % (n) | 17.6 (430) | 22.1 (540) |
| MVPA | | |
| MVPA 0 sessions: % (n) | 19.4 (473) | 24.8 (604) |
| MVPA min/week among active: median (IQR) | 125 (210.0) | 150 (220.0) |
| Mean (SD) | 238.1 (424.0) | 230.9 (398.8) |
| MVPA min/week among entire sample: median (IQR) | 90 (180.0) | 90 (224.5) |
| Mean (SD) | 196.2 (395.4) | 179.8 (364.7) |
| MSA | | |
| MSA: % (n) | | |
| 0 days/week | 55.3 (1557) | 56.2 (1584) |
| 1 day/week | 11.1 (314) | 9.6 (271) |
| 2 days/week | 10.4 (293) | 7.9 (223) |
| 3 days/week | 9.0 (253) | 7.6 (213 |
| 4+ days/week | 7.0 (197) | 11.3 (320) |
| MSA: median (IQR) | 0 (2) | 0 (2) |
| Meet guidelines: % (n) | | |
| Neither MVPA nor MSA | 45.1 (1271) | 44.6 (1256) |
| Meet MVPA (≥150 min/week) only | 21.3 (599) | 21.2 (598) |
| Meet MSA (≥2 days/week) only | 11.7 (330) | 11.9 (336) |
| Meets both MVPA and MSA | 14.5 (410) | 14.8 (416) |
| Change in form of exercise among those who exercise: % (n) | | |
| None of the same | – | 17.1 (331) |
| Some of the same | – | 32.7 (632) |
| About half the same | – | 8.7 (168) |
| Mostly the same | – | 26.0 (503) |
| Exactly the same | – | 15.5 (299) |

*As part of the HEBECO baseline survey, participants were asked about their physical activity levels Before/Since the COVID-19 pandemic has started to affect their livese in any way.
MSA, muscle-strengthening activity; MVPA, moderate-to-vigorous aerobic physical activity.

post-16 years of age (aOR=0.73, 95% CI 0.58 to 0.91) and access to a garden/balcony to exercise comfortably in (aOR=0.75, 95% CI 0.60 to 0.94) were significantly less likely to decrease MVPA activity (table 4). Those with conditions limiting PA (aOR=1.74, 95% CI 1.27 to 2.39), living in total isolation (aOR=3.81, 95% CI 2.33 to 6.23) and experiencing a deterioration in psychological well-being during lockdown (aOR=1.40, 95% CI 1.15 to 1.71) were significantly more likely to decrease MVPA from before COVID-19 levels.

White ethnicity, being employed during COVID-19 and living in total isolation were significantly associated with lower odds of increasing MVPA among this group. At least high school education and living with children were significantly associated with higher odds of increasing MVPA activity.

### RQ4: associations with changes in MSA

Being employed during lockdown (aOR=1.81, 95% CI 1.34 to 2.46) and having overweight (BMI=25–29.99 kg/

m²; aOR=1.88, 95% CI 1.39 to 2.55) or obesity (BMI ≥30 kg/m²; aOR=23.38, 95% CI 2.23 to 5.14) were significantly associated with higher odds of decreasing MSA activity. Older age (35–64 years, aOR=0.22, 95% CI 0.15 to 0.33; and 65+ years, aOR=0.34, 95% CI 0.20 to 0.58) and deterioration in psychological well-being (aOR=0.62, 95% CI 0.46 to 0.83) were significantly associated with lower odds of increasing MSA. See table 5 for details.

### DISCUSSION

This study shows that the first COVID-19 lockdown in the UK affected the levels of MVPA and MSA differently, with several unique factors associated with engagement in and changes in these two activity types. Furthermore, although MVPA and MSA declined among a substantial proportion of UK adults during the first lockdown, which is in line with a majority of studies reporting declines across different measures of physical activity,[17 20] the

**Table 3** Predictors of meeting both, MVPA only and MSA only WHO guidelines since COVID-19 in comparison to not meeting WHO guidelines for MVPA and MSA (reference, n=1178)

| | Both MVPA and MSA (n=388) | | | MVPA only (n=570) | | | MSA only (n=306) | | |
|---|---|---|---|---|---|---|---|---|---|
| | aOR | 95% CI | P value | aOR | 95% CI | P value | aOR | 95% CI | P value |
| Female | 1.08 | 0.82 to 1.41 | 0.588 | 0.79 | 0.62 to 1 | 0.049 | 1.02 | 0.77 to 1.35 | 0.906 |
| Age ≤34 years | | | | | | | | | |
| Age: 35–64 years | 0.7 | 0.49 to 0.99 | 0.047 | 1.11 | 0.78 to 1.56 | 0.566 | **0.56** | **0.38 to 0.82** | **0.003** |
| Age: 65+ years | **0.53** | **0.32 to 0.87** | **0.012** | 1.1 | 0.71 to 1.7 | 0.676 | 0.99 | 0.6 to 1.61 | 0.961 |
| White ethnicity | 1.61 | 1.03 to 2.52 | 0.036 | 1.57 | 0.99 to 2.49 | 0.054 | 0.72 | 0.47 to 1.1 | 0.13 |
| Household income ≥50 000 GBP | | | | | | | | | |
| Income: <50 000 GBP | **0.65** | **0.46 to 0.91** | **0.011** | 0.71 | 0.52 to 0.96 | 0.028 | 1.13 | 0.76 to 1.67 | 0.556 |
| Income: prefer not to say | **0.44** | **0.25 to 0.77** | **0.004** | 0.63 | 0.38 to 1.03 | 0.065 | 0.88 | 0.49 to 1.59 | 0.666 |
| Education: post-16 years of age or higher | **1.57** | **1.14 to 2.17** | **0.006** | 1.04 | 0.81 to 1.35 | 0.745 | **1.74** | **1.24 to 2.45** | **0.002** |
| Employed | 0.76 | 0.55 to 1.05 | 0.094 | 1.09 | 0.82 to 1.43 | 0.562 | 1.06 | 0.75 to 1.49 | 0.734 |
| Laid-off/furloughed | 0.79 | 0.51 to 1.23 | 0.295 | 1.05 | 0.71 to 1.54 | 0.813 | 1.14 | 0.71 to 1.83 | 0.584 |
| Condition limiting physical activity | **0.44** | **0.28 to 0.71** | **0.001** | **0.43** | **0.29 to 0.63** | **<0.001** | 0.97 | 0.65 to 1.45 | 0.881 |
| BMI ≤24 kg/m$^2$ | | | | | | | | | |
| BMI: 25–29.99 kg/m$^2$ | 0.81 | 0.6 to 1.1 | 0.176 | 1.01 | 0.78 to 1.32 | 0.926 | 0.72 | 0.53 to 0.99 | 0.043 |
| BMI: obese: 30+ kg/m$^2$ | **0.49** | **0.33 to 0.73** | **<0.001** | 0.87 | 0.64 to 1.19 | 0.394 | **0.37** | **0.24 to 0.56** | **<0.001** |
| Total isolation | **0.44** | **0.23 to 0.83** | **0.011** | **0.26** | **0.14 to 0.5** | **<0.001** | 0.83 | 0.49 to 1.4 | 0.478 |
| Minor/no COVID-19 risk percept | 1 | 0.74 to 1.34 | 0.977 | 1.2 | 0.93 to 1.55 | 0.167 | 0.94 | 0.68 to 1.3 | 0.718 |
| Living with children | 0.81 | 0.57 to 1.16 | 0.254 | 0.75 | 0.55 to 1.03 | 0.079 | 0.66 | 0.44 to 0.99 | 0.045 |
| Living with vulnerable | 1.17 | 0.82 to 1.68 | 0.395 | 0.86 | 0.61 to 1.2 | 0.366 | 0.87 | 0.58 to 1.3 | 0.499 |
| Access garden/balcony to exercise comfortably in | 1.1 | 0.81 to 1.51 | 0.533 | 1.02 | 0.79 to 1.33 | 0.861 | 0.8 | 0.58 to 1.09 | 0.159 |
| Access green space within walking distance | 1.27 | 0.96 to 1.69 | 0.096 | 1.31 | 1.03 to 1.66 | 0.028 | 0.91 | 0.68 to 1.23 | 0.548 |
| Smoker | 0.72 | 0.51 to 1.02 | 0.062 | 0.93 | 0.7 to 1.22 | 0.593 | 0.8 | 0.57 to 1.14 | 0.218 |
| Weekly alcohol drinking, none (reference) | | | | | | | | | |
| Weekly or less | 1 | 0.68 to 1.49 | 0.988 | 1.43 | 1.02 to 2 | 0.038 | 1.06 | 0.71 to 1.59 | 0.782 |
| More than weekly | 1.04 | 0.73 to 1.5 | 0.811 | 1.28 | 0.94 to 1.74 | 0.121 | 0.89 | 0.61 to 1.29 | 0.528 |
| Deteriorated psychological well-being | **0.56** | **0.43 to 0.73** | **<0.001** | **0.71** | **0.56 to 0.89** | **0.003** | 0.72 | 0.55 to 0.96 | 0.025 |
| Meeting WHO PA* recommendations before COVID-19 (meeting neither=reference) | | | | | | | | | |
| Meeting both before | **16.11** | **11.24 to 23.07** | **<0.001** | **2.31** | **1.58 to 3.39** | **<0.001** | **3.72** | **2.45 to 5.62** | **<0.001** |
| Meeting only MVPA before | **3.88** | **2.64 to 5.7** | **<0.001** | **7.57** | **5.82 to 9.84** | **<0.001** | 1.47 | 0.94 to 2.3 | 0.092 |
| Meeting only MSA before | **6.38** | **4.26 to 9.54** | **<0.001** | 0.74 | 0.44 to 1.25 | 0.264 | **9.61** | **6.7 to 13.77** | **<0.001** |

Results from fully adjusted models using all variables listed on weighted data with Benjamini-Hochberg false discovery rate (BH FDR) correction of p values (significant in bold).

All models were also adjusted for the time of enrolment and time when COVID-19 started to affect individuals in any way. Following the BH FDR correction, the highest p value that met the threshold for significance was p=0.019.

*Meeting WHO's physical activity recommendations before the first COVID-19 lockdown in the UK.

aOR, adjusted OR; BMI, body mass index; MSA, muscle-strengthening activity; MVPA, moderate-to-vigorous aerobic physical activity.

**Table 4** Independent associations of change in MVPA (models fully adjusted using all the variables listed)

| | Sample active (≥30 min/week) Before COVID-19 (weighted n=1857) Decrease (n=868) versus not | | | Sample inactive (<30 min/week) Before COVID-19 (weighted n=585) Increase (n=174) versus not | | |
|---|---|---|---|---|---|---|
| | aOR | 95% CI | P value | aOR | 95% CI | P value |
| Female | 1.22 | 1 to 1.49 | 0.047 | 1.56 | 1.02 to 2.38 | 0.04 |
| Age (in decades) | **0.91** | **0.85 to 0.98** | **0.015** | **0.72** | **0.61 to 0.84** | **<0.001** |
| White ethnicity | **0.62** | **0.44 to 0.86** | **0.005** | 1.01 | 0.44 to 2.33 | 0.982 |
| Household income ≥50 000 GBP | | | | | | |
| Income:<50 000 GBP | 1.3 | 1 to 1.67 | 0.048 | 0.49 | 0.26 to 0.93 | 0.03 |
| Income: prefer not to say | 1.22 | 0.81 to 1.85 | 0.335 | 0.36 | 0.14 to 0.93 | 0.034 |
| Education: post-16 years of age or higher | **0.73** | **0.58 to 0.91** | **0.006** | **2.26** | **1.4 to 3.63** | **0.001** |
| Employed | 0.86 | 0.69 to 1.07 | 0.18 | **0.51** | **0.31 to 0.85** | **0.009** |
| Laid-off/furloughed | 1.05 | 0.76 to 1.47 | 0.75 | 1.3 | 0.72 to 2.37 | 0.385 |
| Condition limiting PA | **1.74** | **1.27 to 2.39** | **0.001** | 0.49 | 0.27 to 0.89 | 0.019 |
| BMI ≤24 kg/m$^2$ | | | | | | |
| BMI: 25–29.99 kg/m$^2$ | 1.03 | 0.82 to 1.28 | 0.816 | 1.54 | 0.94 to 2.53 | 0.088 |
| BMI: obese: 30+ kg/m$^2$ | 0.87 | 0.66 to 1.14 | 0.317 | 0.74 | 0.42 to 1.31 | 0.307 |
| Total isolation | **3.81** | **2.33 to 6.23** | **<0.001** | **0.29** | **0.12 to 0.73** | **0.008** |
| Minor/no COVID-19 risk percept | 0.99 | 0.8 to 1.23 | 0.926 | 1.2 | 0.76 to 1.92 | 0.436 |
| Living with children | 0.84 | 0.65 to 1.09 | 0.187 | **2.32** | **1.27 to 4.24** | **0.006** |
| Living with vulnerable | 0.78 | 0.59 to 1.02 | 0.074 | 0.9 | 0.51 to 1.59 | 0.716 |
| Access garden/balcony to exercise comfortably in | **0.75** | **0.6 to 0.94** | **0.011** | 0.72 | 0.46 to 1.13 | 0.157 |
| Access green space within walking distance | 0.82 | 0.67 to 1.01 | 0.059 | 1.1 | 0.72 to 1.67 | 0.665 |
| Smoker | 1.09 | 0.85 to 1.4 | 0.504 | 1.15 | 0.73 to 1.8 | 0.551 |
| Weekly alcohol drinking, none (reference) | | | | | | |
| Weekly or less | 0.78 | 0.59 to 1.04 | 0.087 | 0.77 | 0.43 to 1.38 | 0.378 |
| More than weekly | 0.78 | 0.6 to 1.02 | 0.068 | 0.88 | 0.52 to 1.5 | 0.644 |
| Deteriorated psychological well-being | **1.4** | **1.15 to 1.71** | **0.001** | 1.09 | 0.72 to 1.66 | 0.682 |

All models were also adjusted for the time of enrolment and time when COVID-19 started to affect individuals in any way. Following the Benjamini-Hochberg false discovery rate (BH FDR) correction, the highest p value that met the threshold for significance was p=0.019 (the significant results are presented in bold).
Segmented analyses of active sample (before COVID-19; MVPA activity ≥30 min; predicting decrease by ≥20 min) and inactive sample (before COVID-19 MVPA; activity <30 min; predicting increase by ≥20 min).
aOR, adjusted OR; BMI, body mass index; MVPA, moderate-to-vigorous aerobic physical activity.

present findings show that a substantial minority of UK adults maintained or increased MVPA or MSA levels. These findings underline the importance of measuring MVPA and MSA, and their correlates, separately.

### Meeting WHO recommendations for MVPA and MSA before and since COVID-19

Adherence to WHO recommendations for both MVPA and MSA before and since COVID-19 was low—at about 15%, which is lower than 26% reported in the Health Survey for England (HSE) for 2012–2016.[40] Also, in comparison to the findings from HSE (36%), fewer of the adults in this study (21%) met MVPA only recommendations, but more met MSA only recommendations (12% in this study vs 1% in HSE). The current MSA levels are

reflecting other epidemiological data.[9 41] The differences between the HEBECO and HSE findings could be due to the differences in the measures of MSA and MVPA used, participant characteristics (eg, in comparison to the HSE study,[40] the HEBECO sample included more educated adults who may be more aware of the benefits of MSA), or reflect genuine changes to the patterns of physical activity in the UK across time. In line with other studies,[40 42] older adults (aged 65+ years), adults who were obese, those with before COVID-19 household income below 50 000 GBP and those who had conditions limiting physical activity were at greater risk of not meeting the WHO recommendations for both MVPA and MSA. These groups should be targeted by future interventions aimed at increasing

**Table 5** Independent associations of change in MSA (models fully adjusted using all the variables listed)

| | Sample active (≥1 day/week) | | | Sample inactive (0 days/week) | | |
|---|---|---|---|---|---|---|
| | Before COVID-19 (weighted n=985) | | | Before COVID-19 (weighted n=1456) | | |
| | Decrease (n=444) versus not | | | Increase (n=239) versus not | | |
| | aOR | 95% CI | P value | aOR | 95% CI | P value |
| Female | 0.94 | 0.71 to 1.23 | 0.634 | 0.96 | 0.7 to 1.31 | 0.792 |
| Age (in decades) | 1.03 | 0.94 to 1.14 | 0.516 | – | – | – |
| Age ≤34 years | | | | | | |
| Age: 35–64 years | – | | – | **0.22** | **0.15 to 0.33** | **<0.001** |
| Age: 65+ years | – | | – | **0.34** | **0.2 to 0.58** | **<0.001** |
| White ethnicity | 0.75 | 0.51 to 1.11 | 0.154 | 0.71 | 0.41 to 1.23 | 0.216 |
| Household income ≥50 000 GBP | | | | | | |
| Income:<50 000 GBP | 1.11 | 0.8 to 1.55 | 0.535 | 0.67 | 0.45 to 0.99 | 0.047 |
| Income: prefer not to say | 0.94 | 0.54 to 1.62 | 0.818 | 0.56 | 0.29 to 1.08 | 0.084 |
| Education: post-16 years of age or higher | 0.82 | 0.59 to 1.15 | 0.26 | 1.2 | 0.85 to 1.7 | 0.31 |
| **Employed** | **1.81** | **1.34 to 2.46** | **<0.001** | 1.08 | 0.75 to 1.55 | 0.671 |
| Laid-off/f | 1.34 | 0.85 to 2.12 | 0.205 | 1.3 | 0.8 to 2.1 | 0.289 |
| Condition limiting physical activity | 1.11 | 0.72 to 1.72 | 0.632 | 0.64 | 0.38 to 1.05 | 0.078 |
| BMI ≤24.99 kg/m$^2$ | | | | | | |
| BMI: 25–29.99 kg/m$^2$ | **1.88** | **1.39 to 2.55** | **<0.001** | 0.89 | 0.63 to 1.26 | 0.512 |
| BMI: obese: 30+ kg/m$^2$ | **3.38** | **2.23 to 5.14** | **<0.001** | 0.64 | 0.42 to 0.97 | 0.037 |
| Total isolation | 1.87 | 1.02 to 3.42 | 0.042 | 0.8 | 0.42 to 1.52 | 0.499 |
| Minor/no COVID-19 risk percept | **1.43** | **1.06 to 1.92** | **0.019** | 0.93 | 0.65 to 1.31 | 0.669 |
| Living with children | 1.01 | 0.71 to 1.44 | 0.964 | 1.48 | 0.94 to 2.31 | 0.088 |
| Living with vulnerable | 1.17 | 0.79 to 1.74 | 0.437 | 0.95 | 0.63 to 1.44 | 0.809 |
| Access garden/balcony to exercise comfortably in | 1.02 | 0.74 to 1.41 | 0.891 | 0.84 | 0.6 to 1.17 | 0.301 |
| Access green space within walking distance | 1.07 | 0.79 to 1.43 | 0.667 | 0.87 | 0.64 to 1.18 | 0.37 |
| **Smoker** | **0.63** | **0.43 to 0.91** | **0.013** | 1.04 | 0.74 to 1.46 | 0.828 |
| Weekly alcohol drinking, none (reference) | | | | | | |
| Weekly or less | 1.11 | 0.74 to 1.65 | 0.62 | 1.13 | 0.73 to 1.76 | 0.58 |
| More than weekly | 1.26 | 0.86 to 1.85 | 0.237 | 1.06 | 0.71 to 1.59 | 0.786 |
| Deteriorated psychological well-being | 1.27 | 0.97 to 1.66 | 0.082 | **0.62** | **0.46 to 0.83** | **0.002** |

All models were also adjusted for the time of enrolment and time when COVID-19 started to affect individuals in any way. Following the Benjamini-Hochberg false discovery rate (BH FDR) correction, the highest p value that met the threshold for significance for was p=0.019. Segmented analyses of active sample (before COVID-19; MSA activity ≥1 day/week; predicting decrease by ≥1 day/week) and less inactive sample (before COVID-19; MSA activity 0 days/week; predicting increase by ≥1 day/week). Findings from fully adjusted logistic regression models on weighted data and using BH FDR adjustment (significant in bold).
aOR, adjusted OR; BMI, body mass index; MSA, muscle-strengthening activity.

physical activity. Such targeting could include dedicated reach out campaigns of more generic interventions among these groups, as well as developing interventions that address the unique circumstances in which these populations can engage in physical activity.

Meeting the recommendations for MVPA and MSA before COVID-19 was strongly predictive of meeting them since COVID-19. Additionally, adults who remained active tended to maintain at least similar levels and forms of the exercises as in the pre-pandemic period. Thus, as UK adults remain relatively consistent in their behaviour,

even during the lockdown, this study highlights the need to especially support those with low activity levels to develop healthy exercising routines.

### Changes to MVPA and MSA levels from before to since the COVID-19 pandemic start

Although group level data suggest little change in physical activity from before to since COVID-19, especially in MSA, about a third of adults decreased and another third increased their MVPA from before to since COVID-19. Those who changed MVPA decreased or increased their

MVPA levels by over 3 hours/week and 2 hours/week, respectively. Those who changed MSA either gained or lost 2 days of MSA per week. These changes are substantial and likely to affect health outcomes.

Decreases in MVPA were found among other adults samples in the UK,[19] Italy[26] and Spain.[27] However, the finding that over 70% of UK adults maintained or increased their MVPA could be at least partially attributed to the UK government's consistent lockdown policies that allowed leaving the house for exercising, which is in contrast to other countries with more restrictive policies.[27 43]

The maintenance of MSA levels could be at least partially explained by the low before COVID-19 MSA levels. Additionally, many MSA exercises can be performed with no equipment and at one's home (eg, pilates and push-ups) and thus may be less affected by social distancing measures. This is further supported by the finding that total isolation (ie, not leaving the house for any reason) was associated with MVPA but not MSA levels.

### Factors associated with changes and maintenance of MVPA and MSA since COVID-19

Older adults were more likely to maintain their before COVID-19 MVPA and MSA levels, including inactivity. Due to more established routines and lower baseline physical activity, this group might have been less affected by the lockdown restrictions, such as gym closures or team sports bans.[44]

MVPA inactivity was more commonly maintained by those who were employed or living in total isolation. Such adults likely had fewer opportunities or flexibility to exercise. Additionally, those who had a health condition limiting physical activity and those living in total isolation tended to decrease MVPA. Factors that were associated with maintaining MVPA activity since COVID-19 were white ethnicity, higher education and having a garden or balcony large enough to exercise.

Increasing MVPA levels among those who were previously inactive was most common among those with higher education, which is in line with prior studies,[45] and those who lived with children. Caring for children might have promoted some forms of MVPA, including outdoors, particularly as parents became responsible for children's physical education during school lockdown closures.

In terms of changes in MSA levels, older adults were at a higher risk of maintaining MSA inactivity, while adults with higher BMI levels were at a greater risk of decreasing MSA from before to since COVID-19. Adults with higher BMI reported lower levels of physical activity during lockdowns in other studies as well.[28] Therefore, both older adults and those with higher BMI should be among priority groups targeted with MSA interventions, particularly as MSA can bring important clinical benefits to these two groups.[46–49] Other priority groups for MSA training are those with lower household income and lower educational attainment.

Finally, as found previously,[24] psychological well-being was associated with physical activity levels. In the present study, deterioration in psychological well-being was associated with not meeting guidelines for MVPA or decreasing MVPA levels, but not with MSA. Due to the cross-sectional design, causality cannot be assumed as poorer psychological well-being (eg, lower mood, anxiety and stress) can be both a predictor and a consequence of low exercise levels.[50 51] However, interventions aimed at improving physical activity are likely to improve mental health as well.[50–52]

### Strengths and limitations

The study is one of a few that assessed the levels of, and changes in, both MVPA and MSA during the first UK lockdown. The study drew on previously used measures of MVPA and MSA,[53] which were supplemented by images to clarify exercise types. This study also benefits from measuring and including in the models a large number of correlates and covariates that could act as potential confounding variables. Several sensitivity analyses were conducted to test the robustness of findings. The key limitations are that this was a cross-sectional study among a self-selected sample that relied on self-report and recall that are prone to bias. This could lead to over-reporting of physical activity and the possibility that the wider population had an even poorer physical activity profile during the pandemic. A small sample (11%) of all HEBECO baseline participants were excluded from the present analyses due to incomplete data (eg, attrition before reporting physical activity). However, while some of the characteristics of the excluded participants could be associated with greater physical activity (ie, younger age, non-female gender and lower BMI), others may be associated with lower physical activity levels (ie, not being in employment, not having access to garden or balcony, being a smoker or being of non-white ethnicity). On balance, the present findings, therefore, may be reflective of the changes in physical activity at the population level in the UK. Finally, the unfolding of the COVID-19 pandemic has coincided with the seasonal change from winter to spring. Without a true baseline from the same period in 2019, it is not possible to tease apart the effect of weather change from that of the pandemic.

### Implications

This study adds to the growing body of literature that emphasises the importance of researching MSA in addition to the more commonly assessed MVPA.[9] It also shows that although the first UK lockdown adversely affected many individuals, a considerable proportion of the adults managed to maintain or even improve MVPA and MSA. These findings suggest that future interventions and policies should not only aim to prevent deterioration in physical activity but also try to capitalise on the opportunities brought by lifestyle disruption to support increases in activity levels. We have at our disposal many interventions and tools, including those that are digital based or

technology based, which could be offered at relatively low cost to large target populations while adhering to even the strictest lockdown measures. Efforts should focus now on identifying acceptable and effective ways of delivering such interventions for both MVPA and MSA, especially among the groups in the population that are at risk of poorer PA outcomes and given the ongoing challenges brought by the COVID-19 pandemic.

## CONCLUSIONS

The findings highlight social inequalities in how the first lockdown in the UK has affected physical activity levels, with differential impact on aerobic and strength training. Dedicated interventions are needed to support MVPA and MSA, especially among those who are older, have lower income and have higher BMI with general low activity levels.

**Acknowledgements** We are grateful to all participants who have been supporting our research. We would like to thank Public Health England, and particularly members of the Behavioural Insights at Public Health England for providing feedback on the survey wording. We would like to thank Public Health England, Cancer Research UK, local authorities, Mayors' offices, as well as charities and other organisations in the UK, including the Asthma UK and British Lung Foundation Partnership, for supporting our recruitment campaign.

**Contributors** AH conceived the idea in consultation with PL, AF and LS, and drafted the first version of the manuscript. AH and VS analysed the data. All authors (AH, VS, AF, DK, LS and PL) interpreted the data, critically revised the manuscript and approved the final draft before submission. AH is the guarantor.

**Funding** This project is partially funded by an ongoing Cancer Research UK programme grant to UCL Tobacco and Alcohol Research Group (C1417/A22962) and by SPECTRUM, a UK Prevention Research Partnership Consortium (MR/S037519/1). PL's salary is paid for by a Yorkshire Cancer Research grant (UCL420) and a Cancer Research UK project grant (C43975/A27498).

**Competing interests** None declared.

**Patient and public involvement** Patients and/or the public were not involved in the design, or conduct, or reporting, or dissemination plans of this research.

**Patient consent for publication** Not applicable.

**Ethics approval** This study involves human participants. The study was approved by the by UCL Research Ethics Committee at the UCL Division of Psychology and Language Sciences (CEHP/2020/579). Participants gave informed consent to participate in the study before taking part.

**Provenance and peer review** Not commissioned; externally peer reviewed.

**Data availability statement** Data are available upon reasonable request. Data is stored securely at UCL. Interested researchers can contact the HEealth BEhaviours during the COVID-19 pandemic study leads for any inquiries regarding study data and collaboration (Dr Aleksandra Herbec, a.herbec@ucl.ac.uk, and Professor Lion Shahab, lion.shahab@ucl.ac.uk).

**ORCID iD**
Aleksandra Herbec http://orcid.org/0000-0002-3339-7214

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
