## [Reviewer comments · BMJ Open]

ARTICLE DETAILS

TITLE (PROVISIONAL)	Correlates of and changes in aerobic physical activity and strength training before and after the onset of COVID-19 pandemic in the UK – findings from the HEBECO study.
AUTHORS	Herbec, Aleksandra; Schneider, Verena; Fisher, Abigail; Kale, Dimitra; Shahab, Lion; Lally, Phillippa

VERSION 1 – REVIEW

REVIEWER	Mike Trott Anglia Ruskin University - Cambridge Campus
REVIEW RETURNED	21-Oct-2021

GENERAL COMMENTS	This paper examines the differences in MVPA and MSA from pre COVID to June, during the first COVID19 lockdown. Overall the paper is novel and the results interesting - most of my comments are minor in nature. One general comment is the quality of grammar in the writing. There are several sentences and paragraphs in the manuscript that need editing regarding the use of tense and sentence structure. I have highlighted some of these in the manuscript but they became so numerous that I recommend any revisions to be fully checked for quality of English in the writing style. ABSTRACT - Results - why are two types of units used when reporting the results? In line 24 min/week is used and then days/week. Please, if possible keep it consistent. At the end of the results section the authors do not provide effect sizes. Because the abstract is the first, and sometimes only thing, people read, please provide effect sizes. - Conclusion - I feel this sentence doesn't add anything at all to the abstract - what are the implications? INTRODUCTION - lines 14-18. I am curious as to why the authors decided to use the WHO recommendations rather than the UK (2019) ones? Especially given it was a UK sample? Line 30 - please make this clear that things like the furlough scheme potentially allowed people more time to exercise, rather than 'facilitate' - its a bit vague. Line 53 - it is curious that you use subjective measures of PA as a limitation of past studies here when your study uses subjective measurement also? Moreover, few studies that assessed should be 'of the few studies' (add 'of the'). METHODS P7 line 57 'creased' type - do you mean created? P7 line 58 - all these should be in the past tense P8 line 21/22 - why was a cut off point of 50k used? is this based on any data, such as median household income, or was this just made up? Also, please keep the formatting of the numbers consistent,
--

	here you have a space (50 000), whereas in other places you (incorrectly) used a decimal point. P8 line 56 please add a ref for SPSS P9 line 8/9 - it isnt 100% what was used as covariates in the regression - you mean EVERYTHING you measured was chucked into a model? Why was this done and not a stepwise regression, for example? P9 line 26 present should be presents P9 line 31. The term 'and have not changed the conclusions' when talking about sensitivity analyses is rather vague. I assume you mean did not change the magnitude, significance, or direction of results? Please just be a little bit more explicit. RESULTS - ALL When reporting the results. please include the effect sizes and 95% CIs - although they have been provided in the table, it would make sense to include them in the written results as well, that way the reader does not have to flit between the table and the written prose to get the key information. P11 line 21: target should be targeted. Please also add - such as what? What kind of targeted interventions do you as the author recommend? P12 line 34 should read strengths and limitations (add the s's) P12 line 40 - being measured to control for confounding...this makes no sense - i assume you mean being measured to control for confounding [variables]. line 46 should be excluded not exclude.
--	--

VERSION 1 – AUTHOR RESPONSE

Reviewer: 1

Dr. Mike Trott, Anglia Ruskin University - Cambridge Campus

Comments to the Author:

This paper examines the differences in MVPA and MSA from pre COVID to June, during the first COVID019 lockdown. Overall the paper is novel and the results interesting - most of my comments are minor in nature. One general comment is the quality of grammar in the writing. There are several sentences and paragraphs in the manuscript that need editing regarding the use of tense and sentence structure. I have highlighted some of these in the manuscript but they became so numerous that I recommend any revisions to be fully checked for quality of English in the writing style.

Response: Thank you for reviewing article and for providing comments. We have responded to them (below) and also revised the text to improve grammar and the quality of English.

ABSTRACT

1. Results - why are two types of units used when reporting the results? In line 24 min/week is used and then days/week. Please, if possible keep it consistent.

Response 1: We report two types of units because the two exercise types were assessed using different units (as per WHO guidance) and we cannot unify them. MVPA was measured in minutes/week, and MSA as days/week (Bennie et al, 20198).

2. At the end of the results section the authors do not provide effect sizes. Because the abstract is the first, and sometimes only thing, people read, please provide effect sizes.

Response 2: We added the effect sizes and more details to the abstract: “The odds for decreasing MVPA were lower for white ethnicity (aOR=0.62,0.44-.86), post-16 years of age education (aOR=0.73,0.58-0.91), access to garden/balcony (aOR=0.75,0.60-0.94), and were higher for those living in total isolation (aOR=3.81,2.33-6.23), with deteriorated psychological wellbeing

(aOR=1.40,1.15-1.71) and conditions limiting physical activity (aOR=1.74,1.27-2.39). The odds for decreasing MSA were higher for having overweight (aOR=1.88,1.39-2.55), obesity (aOR=23.38,2.23-5.14) and being employed (aOR=1.81,1.34-2.46).”

3. Conclusion - I feel this sentence doesn't add anything at all to the abstract - what are the implications? Response 3: We added a sentence on implications:

“Conclusion: Aerobic and strength training were differently impacted during the first UK lockdown, with poorer outcomes associated with older age, lower education, and higher body mass index. Targeted interventions may be required to avoid pandemic-related inequities in physical activity.”

INTRODUCTION

3. - lines 14-18. I am curious as to why the authors decided to use the WHO recommendations rather than the UK (2019) ones? Especially given it was a UK sample?

Response: We selected the most efficient manner to assess clinically relevant criteria for physical activity. The WHO guidelines and the UK (2019) overlap in the key respects that were assessed in this study (i.e. the overall time recommended of 150min of MVPA activity, and a minimum of 2 days/week of MSA). Furthermore, from a research perspective it was important for us to assess indicators that could be compared across studies undertaken in other countries.

Importantly, the UK guidelines

(https://assets.publishing.service.gov.uk/government/uploads/system/uploads/attachment_data/file/832868/uk-chief-medical-officers-physical-activity-guidelines.pdf; but also other country's guidelines) tend to be much more complex and more difficult to score in a survey format. Specifically, they provide additional considerations for the type or frequency of the recommended activity levels and these also tend to vary depending on age or patient groups. The study used data from the HEBECO study that did not solely focus on physical activity, but covered additional health behaviours. Therefore, it was not feasible to assess all the dimensions of exercising to assess adherence to the full UK guideline, especially as the latter would require more of a case-by-case assessment of individual's health, age and fitness status.

We added to the introduction (2nd paragraph):

“Adults who engage in less than 30 minutes of moderate physical activity per week are considered to be inactive^{4,5}. In the UK⁶ and other countries (e.g. the US⁷) guidelines for physical activity frequency and duration are complex, and often vary by age or clinical profile of the target population. However, a consensus exists that for the best health outcomes adults are recommended to engage in moderate-to-vigorous intensity aerobic physical activity (MVPA; i.e. activities that increase heart rate and make one feel warmer) for at least 150 minutes/week, as well as in muscle-strengthening activity (MSA; e.g., strength/resistance training) for at least two days/week⁸. MVPA and MSA lower morbidity and mortality both independently and combined⁹⁻¹⁴. Importantly, improving MVPA or MSA requires different strategies and interventions at both population and individual level, thus both are important outcomes to research¹⁵.”

4. Line 30 - please make this clear that things like the furlough scheme potentially allowed people more time to exercise, rather than 'facilitate' - its a bit vague.

Response: we replaced 'facilitate' with 'enable'.

5. Line 53 - it is curious that you use subjective measures of PA as a limitation of past studies here when your study uses subjective measurement also? Moreover, few studies that assessed should be 'of the few studies' (add 'of the').

Response 5: we added a clarification (below) that we meant that some studies did not assess the level of exercise, but just asked for qualitative comparisons (5th para of the intro, bottom p.4):

“Furthermore, research on how the Covid-19 pandemic has affected physical activity to date often focused on moderate-to-vigorous physical activity (MVPA), did not assess muscle-strengthening activity (MSA), failed to distinguish between MVPA and MSA^{17,24,15,25}, or assessed only perceived, qualitative, changes to physical activity levels with respect to pre-pandemic levels (e.g. participants were asked to report if they exercised ‘more’ or ‘less’ since before the pandemic started rather than providing detailed information about MVPA and MSA responses)^{26,27}.”

METHODS

6. P7 line 57 'creased' type - do you mean created?

Response 6: we corrected this.

7. P7 line 58 - all these should be in the past tense.

Response 7: Dichotomous variables were then created: decreased (vs maintained/increased) and increased (vs maintained/decreased) MVPA and MSA levels. OR Dichotomous variables were then created: a decrease (vs maintained/increased) and increased (vs maintained/decreased) MVPA and MSA levels.

8. P8 line 21/22 - why was a cut off point of 50k used? is this based on any data, such as median household income, or was this just made up? Also, please keep the formatting of the numbers consistent, here you have a space (50 000), whereas in other places you (incorrectly) used a decimal point.

Response 8: We used data that was available in the HEBECO survey and chose the 50k cut-off because it was closer to the c.37k average disposable household income in the UK and represented high income families. We corrected the formatting (removed the full stop).

9. P8 line 56 please add a ref for SPSS

Response 9: we added a reference: IBM Corp. Released 2019. IBM SPSS Statistics for Windows, Version 26.0. Armonk, NY: IBM Corp”.

10. P9 line 8/9 - it isnt 100% what was used as covariates in the regression - you mean EVERYTHING you measured was chucked into a model? Why was this done and not a stepwise regression, for example?

Response 10: The aim of this study was to characterise the physical activity levels and their correlates. All the variables that were considered and ultimately entered into the model were selected from the HEBECO dataset based on prior research findings or theoretical considerations, i.e.,

because they are relevant for physical activity levels or are potential confounders. The analysis for this reason was pre-specified. Please note that in general stepwise regression is considered to be problematic, because it includes/excludes covariates on arbitrarily set statistical cut-offs (e.g., $p < 0.05$ / $p < 0.1$) rather than basing model selection on a theory-informed approach (e.g. see <https://journalofbigdata.springeropen.com/articles/10.1186/s40537-018-0143-6>).

11. P9 line 26 present should be presents Response 11: we corrected this.

12. P9 line 31. The term 'and have not changed the conclusions' when talking about sensitivity analyses is rather vague. I assume you mean did not change the magnitude, significance, or direction of results? Please just be a little bit more explicit.

Response 12: we corrected to: "The results of the sensitivity analyses did not change the direction or magnitude of the results."

RESULTS - ALL

13. When reporting the results. please include the effect sizes and 95% CIs - although they have been provided in the table, it would make sense to include them in the written results as well, that way the reader does not have to flit between the table and the written prose to get the key information.

Response 13: we tried not to repeat information in the text and the tables, but we now added the aORs and 95% CI to the text. The changes are marked in the tracked manuscript.

14. P11 line 21: target should be targeted. Please also add - such as what? What kind of targeted interventions do you as the author recommend?

Response 14: we corrected this (added 'ed'). We also added: "These groups should be targeted by future interventions aimed at increasing physical activity. Such targeting could include dedicated reach out campaigns of more generic interventions among these groups, as well as developing interventions that can take into account the unique circumstances in which these populations can engage in physical activity."

We then added at the end of the implications section (p.12):

"Efforts should focus now on identifying acceptable and effective ways of delivering such interventions for both MVPA and MSA, especially among the groups in the population that are at risk of poorer PA outcomes and given the ongoing challenges brought by the Covid-19 pandemic."

15. P12 line 34 should read strengths and limitations (add the s's) Response 15: we corrected this.

16. P12 line 40 - being measured to control for confounding...this makes no sense - i assume you mean being measured to control for confounding [variables].

Response 16: We corrected this: "This study also benefits from measuring and including in the models a large number of correlates and covariates that could act as potential confounding variables."

17. line 46 should be excluded not exclude. Response 17: we corrected this.

VERSION 2 – REVIEW

REVIEWER	Mike Trott Anglia Ruskin University - Cambridge Campus
REVIEW RETURNED	17-Feb-2022
GENERAL COMMENTS	Dear authors, Thank you for your detailed response and for the updated manuscript. I have no more comments to add.